# Efficient Delivery of P3H4 siRNA and Chlorin e6 by cRGDfK-Installed Polyarginine Nanoparticles for Tumor-Targeting Therapy of Bladder Cancer

**DOI:** 10.3390/pharmaceutics14102149

**Published:** 2022-10-10

**Authors:** Lin Hao, Zhenduo Shi, Yang Dong, Jiangang Chen, Kun Pang, Houguang He, Shaoqi Zhang, Wei Wu, Qianjin Zhang, Conghui Han

**Affiliations:** 1Department of Urology, Xuzhou Central Hospital, Xuzhou 221009, China; 2School of Life Sciences, Jiangsu Normal University, Xuzhou 221116, China; 3Medical College of Soochow University, Suzhou 215123, China; 4Department of Urology, The First People’s Hospital of Nantong City, Nantong 226001, China

**Keywords:** non-muscle invasive bladder cancer, polyarginine peptide, photodynamic therapy, prolyl 3-hydroxylase family member 4, small interfering RNA

## Abstract

Purpose: Prolyl 3-hydroxylase family member 4 (P3H4) is a potent prognostic oncogene in bladder cancer (BC), and the inhibition of P3H4 suppresses BC tumor growth. This study aimed to evaluate the efficiency of P3H4 inhibition for BC tumor therapy via tumor-targeting nanoparticles. Methods and results: A linear polyarginine peptide (R9) was synthesized, azide-modified, and then assembled with cyclic pentapeptide cRGDfK. Chlorin e6 (ce6)-conjugated CH3-R9-RGD nanoparticles were prepared for the delivery of siP3H4 into T24 cells in vitro and BC tumors in vivo. Dynamic light scattering analysis identified that the optimum CH3-R9-RGD@siP3H4 molar ratio was 30/1. CH3-R9-RGD@ce6/siP3H4 nanocomposites decreased P3H4 expression and cell proliferation and promoted reactive oxygen species production, apoptosis, and calreticulin exposure in T24 cells in vitro. In vivo experiments showed that CH3-R9-RGD@ce6/siP3H4 nanocomposites caused pathological changes, suppressed BC tumor growth, promoted caspase 3 expression, and enhanced calreticulin exposure in tumor cells. Conclusions: The tumor-targeting CH3-R9-RGD nanocomposites encapsulating siP3H4 and ce6 might be an alternative therapeutic strategy or intravesical instillation chemotherapy for BC.

## 1. Introduction 

Bladder cancer (BC) is the 12th most common cancer in the world and a major burden on health [1]. The estimated incidence and mortality of BC were 550,000 cases and 200,000 deaths in 2018 [2], and 570,000 cases and 210,000 deaths in 2020 [1]. Most BCs (~80%) are non-muscle-invasive BCs (NMIBC) [3]. The standard care for NMIBC is white light cystoscopy-guided transurethral resection with a high incidence of recurrence [4,5]. Intravesical instillation chemotherapy and immunotherapy following resection are the major clinically available approaches for reducing tumor recurrence [6]. 

Photodynamic therapy (PDT) is a delicate option for BC treatment approved in 1993 [7,8]. PDT is based on the capacity of photosensitizer to convert light energy into excess surrounding reactive oxygen species (ROS) to cause cell apoptosis and necrosis [8]. In addition, PDT is a feasible strategy to trigger immunogenic cell death (ICD) in tumor cells [9,10]. The photosensitizer molecule chlorin e6 (ce6) is widely used in the treatment of malignant tumors because it has several biomedical advantages, including non-toxicity, non-specificity, rapid clearance, bright fluorescence, and a high singlet oxygen yield [11,12]. Ce6-conjugated polymers and self-assembled nanoparticles have been widely used to improve ce6 solubility because ce6 has poor water solubility and rapid clearance and is not conducive to cells [12,13]. 

Recent progress in nanotechnology has developed many nanomaterials to trigger the immunogenicity of tumors to improve the therapeutic efficacy [9,11,13,14,15]. Nanomaterials, including gold, liposomes, polymeric, peptide nanostructures, and albumin-based nanoparticles, make excellent nanotechnological drug-delivery systems in cancer therapy application [15,16]. Notably, the conjugation of active tumor targeting cyclic pentapeptide (Arginine–Glycine–Aspartic acid–d-Phenylalanine–Lysine, cRGDfK) that can specifically bind to αvβ3 integrin in tumor sites improves the therapeutic efficiency of cancer-targeting nanoparticles [11,13,17]. This approach has been introduced to intravesical instillation chemotherapy for BC [18]. Cancer-targeting nanoparticles modified by cell-penetrating and RGD-containing cyclic peptides (i.e., c[RGDfK]) are effective delivery systems for cancer-targeting therapy [13,18,19]. In addition, c(RGDfK)-modified nanoparticles are effective delivery systems of small interfering RNAs (siRNAs) [17,18].

Prolyl 3-hydroxylase family member 4 (P3H4/LEPREL4/SC65) is a member of the collagen Leprecan family proteins, which are critical components of endoplasmic reticulum-resident collagen [20,21]. P3H4 is an autoantigen associated with several types of human diseases, including cancers [22,23]. Our previous studies showed that the expression level of P3H4 was upregulated in BC tumor tissues compared with non-tumor tissues [21,24]. We found that BC patients with high expression levels of P3H4 had lower ratios of disease-free survival and overall survival compared with patients who had low levels of P3H4 [21,24]. Experiments based on both in vitro and in vivo experimental models indicated that the knockdown of P3H4 by siRNA inhibited BC cell proliferation and tumor growth [21]. Based on these findings, we hypothesized that P3H4 is a potential therapeutic target for BC. We supposed that the delivery of siRNA targeting P3H4 (siP3H4) to BC tumors might be an effective BC-targeting therapy. 

Herein, we designed and synthesized a photosensitive delivery system of siP3H4 using cRGDfK-modified nanoparticles. The nanoparticles are assembled using azide-modified polyarginine (R9-N3) and cRGDfK, and are effective for encapsulating ce6 and siP3H4 and the delivery of them to BC tumor sites. The in vitro and in vivo experiments showed that the co-delivery of siP3H4 and ce6 to BC tumor cells using cRGDfK-installed nanoparticles triggered the ICD of tumor cells, inhibited tumor growth in vivo, and enhanced the PDT of BC tumors. This delivery system might be an alternative strategy for BC therapy.

## 2. Materials and Methods

### 2.1. Materials

Amino resin, Fmoc-Arg-pbf-OH, dimethylformamide (DMF), trifluoroacetic acid (TFA), dichloromethane (DCM), N,N-diisopropylethylamine (DIPEA), N,N′–dicyclohexylcarbodiimide (DCC), O-(1H-benzotriazol-1-yl)-N,N,N′,N′-tetramethyluronium hexafluorophosphate (HBTU), methanol, sodium dodecyl sulfate (SDS)-polyacrylamide gel, dimethyl sulfoxide (DMSO), acetonitrile, tris(3-hydroxypropyltriazolylmethyl)amine (THPTA), acrylic acid, copper sulphate (CuSO_4_), sodium ascorbate, formaldehyde, xylene, ethanol, and silver nitrate (AgNO_3_) were obtained from Nanjing Shengxing Biotechnology Co., Ltd. (Nanjing, China). The Cell Counting Kit-8 (CCK8) assay kit was purchased from Sangon (Shanghai, China). The FAM-labeled siP3H4 probe, Dichloro-dihydro-fluorescein diacetate (DCFH-DA) assay kit was purchased from Invitrogen (Carlsbad, CA, USA). Triton X-100, bovine serum albumin (BSA), skimmed milk, H_2_O_2_ solution, paraffin, Tween-20, TRIzol^®^ reagent, 4-6-Diamidino-2-phenylindole (DAPI), Dulbecco’s modified Eagle’s medium (DMEM), fetal bovine serum (FBS), paraformaldehyde, goat serum, and phosphate-buffered saline (PBS) were obtained from Gibco BRL (Gaithersburg, MD, USA). cRGD was purchased from ChinaPeptides Co., Ltd. (Shanghai, China). The Annexin V-FITC Apoptosis Detection Kit was acquired from BD Biosciences (Carlsbad, CA, USA). Chlorpromazine (CPZ), wortmannin, methyl-beta-cyclodextrin (MβCD), azido-PEG2-amine, piperidine, cRGDfK, cRADfk, ce6, N-hydroxysuccinimide (NHS), trypan blue, P3H4 siRNA, and negative control siRNA were obtained from Sigma-Aldrich Co., Ltd. (St. Louis, MO, USA). Cell culture dishes of 6 wells, 12 wells, 24 wells, and 96 wells were obtained from Corning (Corning, NY, USA). Diaminobenzidine (DAB) staining, hematoxylin and eosin (H&E) staining, Tris-buffered saline (TBS), and RIPA lysis buffer were acquired from Solarbio (Beijing, China). The bicinchoninic acid (BCA) protein assay kit was purchased from Pierce (Bonn, Germany). Human BC cell lines T24 and MB49 were obtained from American Type Culture Collection (ATCC, Manassas, VA, USA). The HiScript II Reverse Transcription kit and ChamQ Universal SYBR qPCR master mix were obtained from Vazyme Biotech Co., Ltd. (Nanjing, China). Antibodies of anti-Ki-67, anti-Caspase-3, anti-P3H4, anti-calreticulin (CRT), and anti-glyceraldehyde-3-phosphate dehydrogenase gene (GAPDH) were acquired from Abcam (Cambridge, UK). The enhanced chemiluminescence (ECL) system and polyvinylidene fluoride (PVDF) membranes were obtained from Millipore Corp., (Bedford, MA, USA). BALB/c mice were obtained from the Wuhan Institute of Biological Products Co., Ltd. (Wuhan, China). 

### 2.2. Synthesis and Modification of Polyarginine (R9) and CH3-R9-RGD Polymer

The R9 peptide was prepared using the solid phase peptide synthesis methods. Briefly, the Fmoc removal from resin was carried out with DMF-piperidine (4:1). After cleaning the excess polyarginine solution with DMF, Fmoc-Arg-pbf-OH (3×), HBTU (3×), and DIPEA (6×) were added in DMF for 2 h, for the binding of the first Arg to resin. The couplings were repeated eight times using Fmoc-Arg-pbf-OH as described above. Additional coupling of hydrophobic residue to the linear peptide was performed using DMF/DCM, HBTU (3×), and DIPEA (6×) for 2 h. The peptide-attached resin was then washed using DMF, DCM, and methanol three times, respectively. The cleavage of linear peptide from resin was performed using 83% TFA. The crude linear peptide solution was condensed by rotary evaporation, precipitated with cold ether, repeatedly washed several times, filtered, and vacuum dried. Subsequent dialysis was performed for the linear peptide.

The R9-N3 polymer was synthesized by coupling the R9 polymer (1.2 mmol) to azido-PEG2-amine (1.0 mmol; Figure 1). The reaction was conducted in DMSO with HBTU (1.0 mmol) and DIPEA (4.0 mmol). The mixture was stirred overnight at room temperature. Precipitation in the mixture was removed by filtering. cRGDfK was modified with acetylenic linkages. Conjugation of cRGDfK (0.3 mmol, 181 mg) to acrylic acid (2 mmol) was performed using a DCC/NHS covalent coupling reaction in DMSO, supplementing with DCC (0.4 mmol, 102 mg), NHS (0.2 mmol, 22 mg), and DIPEA (1.0 mmol, 165 μL; Figure 1). The reaction was continued with stirring at room temperature for 4–6 h. The CH3-R9-RGD polymer was synthesized by conjugating R9-N3 (0.01 mmol) to modified cRGDfK (0.01 mmol) using a click reaction in DMSO-water (2:1, *v*/*v*) with sodium ascorbate (200 mg), CuSO_4_ (3.2 mg), and THPTA (50 mg; at 30 °C). The reaction was monitored using the high-performance liquid chromatography (HPLC).

The identification of products was performed using the matrix-assisted laser desorption ionization (MALDI) time-of-flight (TOF) imaging mass spectrometry (IMS) (Agilent, QC, Canada). The purification of products was performed using the reverse phase HPLC, with a gradient system from 0 to 100% acetonitrile and water (0.1% TFA, *v*/*v*; elution time of 1 h, 10.0 mL/min).

### 2.3. Synthesis and Characterization of CH3-R9-RGD@ce6 (Nano@ce6) and CH3-R9-RGD@Ce6/siP3H4 (Nano@ce6/siP3H4) Nanocomposites

The Nano@ce6 polyplex was synthesized by conjugating CH3-R9-RGD polymer (10 mg) to ce6 (2 mg, in DMSO). The Nano@Ce6/siP3H4 nanocomposites were subsequently synthesized by complicating the CH3-R9-RGD polymer with P3H4 siRNA at a constant polymer/siRNA molar ratios (0/1, 10/1, 20/1, 30/1, and 40/1). The complexation was conducted by leaving the mixture at 37 °C for 30 min. The optimum Nano@siP3H4 molar ratio was determined using horizontal gel electrophoresis and the Gel Imager System (Vilber Lourmat, Collegien, France) [25]. Particle morphology was characterized using a JEM-100CX II transmission electron microscope (TEM; Joel, Tokyo, Japan). The hydrodynamic size, Zeta potential, and serum stability of nanocomposites were determined using dynamic light scattering (DLS) by a Zetasizer Nano ZS-ZEN3600 instrument (Malvern Instruments; Malvern Instrument, Inc., London, UK) [26]. Particle size and polydispersity index (PDI) were measured at the set time points (0, 12, 24, and 48 h). Each sample was determined six times and the average value was taken as the determination result.

### 2.4. In Vitro Cell Uptake of Self-Assembled Nanocomposites

T24 cells were seeded in 6-well dishes and incubated in DMEM containing 10% FBS for 12 h at 37 °C. Cells were subsequently incubated with ce6 with and without nanocomposites for 24 h at 37 °C. Then, the FAM-labeled siP3H4 was transfected into T24 cells using the Nano@Ce6/siP3H4 nanocomposites (30/1 molar ratio) and incubated for 2 h. Cells were washed three times with PBS and fixed in formaldehyde (400 μL) for 30 min. The cells were subsequently washed with PBS and stained with DAPI for 3 min. The results were imaged using the laser scanning confocal microscopy (LSCM, at 405 nm; Leica Microsystems CMS GmbH, Wetzlar, Germany).

### 2.5. Detection of Intracellular ROS

T24 cells were seeded in 24-well dishes (4 × 10^4^ cells/well) and incubated for 12 h at 37 °C. The Nano@Ce6/siP3H4 nanocomposites (30/1 molar ratio) were added into cell culture and incubated for 4 h. The generation of intracellular ROS (^1^O_2_) was determined using the fluorogenic probe DCFH-DA according to the recommendations from the manufacturer. Subsequent light stimulation was performed at 660 nm with 0.76 W/cm^2^ for 5 min. LSCM was used to determine the production of ^1^O_2_ in T24 cells.

### 2.6. Study of Endocytosis Mechanism of Self-Assembled Nanocomposites

T24 cells were seeded in 6-well dishes (1 × 10^5^ cells/well) and incubated at 37 °C. When reaching ~60% confluency, cells were treated with the endocytosis-specific inhibitors of CPZ (10 μg/mL), wortmannin (50 nM), MβCD (50 μM), and cRGD (100 μM) [13] at 37 °C for 30 min. Then, the FAM-labeled siP3H4 and Nano@ce6 nanocomposites (30/1 molar ratio) were transfected into T24 cells and incubated for 24 h. Cells were also incubated at 4 °C with and without Nano@Ce6 nanoparticles for 24 h. The intensity of the FAM fluorescence signal in cells was analyzed by flow cytometry (FACScan; Becton Dickinson, San Jose, CA, USA).

### 2.7. In Vitro Phototoxicity

The cytotoxicity of nanoparticles in BC cells was evaluated using a CCK8 assay kit. Cells were seeded in 96-well dishes (6 × 10^3^ cells/well) and incubated for 24 h at 37 °C. Subsequent treatment was carried out by adding self-assembled nanocomposites (Nano@ce6 and Nano@ce6/siP3H4) into cell culture at constant concentrations (0, 25, 50, 100, and 200 μg/mL) for 12 h. Light stimulation was performed by irradiation at 660 nm laser light for 1 min, followed by incubation for another 12 h. Cell viability was determined by measuring the absorbance at 450 nm and the relative cell viability was assessed using the formula of cell viability (%) = 100 × [(OD_Exp_ − OD_CK_)/OD_CK_], where OD_CK_ and OD_Exp_ represent the mean OD values of the control and experimental groups, respectively.

### 2.8. In Vitro Cell Apoptosis

Apoptosis assay was performed using the Annexin V and PI staining. T24 cells were seeded in 6-well dishes (1 × 10^5^ cells/ well) and incubated at 37 °C. When reaching ~70% confluency, cells were treated with nanoparticles (Nano, Nano@ce6, and Nano@ce6/siP3H4) for 12 h, irradiated by 660 nm laser light for 1 min, and then incubated for another 36 h at 37 °C. Subsequently, T24 cells were washed three times with cold PBS. Upon resuspension in Annexin V binding buffer, cells were stained with 5 μL of Annexin V-FITC for 15 min and 5 μL of PI for 15 min in the dark, respectively. The assessment of apoptotic (Annexin V-positive) cells was counted by flow cytometry (Becton Dickinson).

### 2.9. Establishment of Bladder Tumor Model

Animal experiments were conducted according to the National Institutes of Health guidelines after obtaining the approval from the Animal Care and Use Committee of Xuzhou Central Hospital (Xuzhou, China; XZXY-LJ-20200416-017). Four- to six-week-old BALB/c mice (22–25 g) were used for the establishment of an animal model. Ninety mice were maintained in standard conditions (55–60% relative humidity, room temperature (18–30 °C), four mice per cage, natural light/dark cycle) with free access to food and water. The in vivo bladder tumor model was established using intravesical inoculation of MB49 BC cells [27,28]. Briefly, bladder epithelium was eroded using AgNO3 (5 µL, 0.3 M, for 15 s) and washed with PBS through a catheter. Cell suspension was inoculated intravesically (1 × 10^6^ cells) and retained for 1 h. At 7 days post cell inoculation, mice were randomly divided into 7 groups with tail vein injection of PBS (50 μL, Control; *n* = 12) and self-assembled nanocomposites (50 μL, 15 mM; Nano@ce6/siNC, Nano@siP3H4, and Nano@siP3H4; 30/1 molar ratio; *n* = 24 in each group) and were repeated every 5 days. At 6 h after each injection, half the mice in experimental groups (*n* = 12) were lightly anesthetized and tumors were irradiated at 660 nm laser light (0.45 W/cm^2^) for 5 min. The qualitative real-time fluorescence imaging was performed to evaluate the in vivo tissue distribution of Nano@ce6/siP3H4 nanoparticles at 2, 12, 24, and 48 h after injection at 7 days post MB49 cell inoculation. The time-dependent in vivo biodistribution of nanoparticles (Nano@ce6/siP3H4) was monitored using an in vivo imaging system (Bruker In-Vivo FX Pro; Bruker, Billerica, MA, USA). Tumor volume was evaluated every other day, and mice were killed on day 20 post MB49 cell inoculation. Tissues of the lung, spleen, heart, small intestine, liver, and kidney were harvested and rinsed, and ex vivo fluorescence images of major organs and tumors were obtained using the Bruker In-Vivo FX Pro imaging system. All tissues were prepared for the Western blot analysis, H&E staining, and immunohistochemical analysis.

### 2.10. Histopathological and Immunohistochemical Analysis

The histopathological change and expression of Ki-67 in tumors were evaluated using H&E staining and immunohistochemical analysis, respectively. Briefly, tumor tissues were fixed in paraformaldehyde (4%), paraffin-embedded, deparaffinized, and hydrated in a series of xylene-ethanol. The 5 μm-thick section was stained with H&E. For immunohistochemical analysis, sections were pretreated in H_2_O_2_ solution and goat serum. Subsequent incubations with specific primary antibody anti-Ki-67 (1:200) and secondary antibody were performed following the standard procedures. DAB staining was used for the evaluation of protein expression.

### 2.11. Immunofluorescence of CRT in BC Cells

CRT expression in BC cells was determined using the immunofluorescence analysis. Briefly, T24 cells or ex vivo tumor cells growing on glass coverslips were fixed with 4% paraformaldehyde and permeabilized with Triton X-100 (0.2% in PBS). Subsequent blocking was performed with BSA (5%). The immunofluorescence analysis of CRT was carried out using the specific primary antibody anti-CRT, Cy3-labeled goat anti-rabbit fluorescent secondary antibody, and DAPI. CRT expression was determined using LSCM. The intensity of Cy3 fluorescence signal in cells was analyzed by flow cytometry (Becton Dickinson).

### 2.12. Quantitative Real-Time PCR (qRT-PCR) Analysis of Genes

The extraction of total RNA from tumor tissues and T24 cells was performed using the TRIzol^®^ reagent. The reverse transcription from RNA into the first-strand cDNA was conducted using a HiScript II Reverse Transcription kit. The qRT-PCR analysis of the P3H4 gene mRNA was performed using a ChamQ Universal SYBR QPCR Master Mix kit and a qTower3 RT-PCR cycler (Analytik Jena, Jena, Germany). The reaction was carried out using the following conditions: 95 °C for 30 s (one cycle); 95 °C for 10 s, 60 °C for 20 s, and 72 °C for 1 min (40 cycles). The relative expression level of P3H4 mRNA was evaluated using the 2^−^^ΔΔ^^Ct^ methods by referring to the Ct values of the GAPDH (internal reference gene).

### 2.13. Western Blot Analysis

The expression of proteins P3H4, CRT, and Caspase 3 in tumor tissues and T24 cells was determined using Western blot analysis. Samples were lysed with RIPA lysis buffer following the manufacturers’ instructions. Protein quantification was assessed using the Pierce BCA kit. SDS-polyacrylamide gel electrophoresis (PAGE, 10%) of proteins was performed at 80 V and 100 mA for 50 min. Proteins were transferred onto PVDF membranes, which were then incubated in 5% skimmed milk in TBS containing 0.1% Tween 20 (TBST) for blocking (at room temperature for 1 h). After washing three times in TBST, membranes were then incubated in solutions containing specific primary antibodies anti-P3H4, anti-CRT, anti-Caspase 3, and anti-GAPDH (1:1000) overnight at 4 °C. After shaking on a shaker for 30 min and washing three times in TBST, subsequent incubation of membranes was performed using horseradish peroxidase-conjugated secondary antibody (1:5000) for 1 h at room temperature. Analysis of protein expression was assessed using the ECL system and Image-Pro Plus 6.0 software (Media Cybernetics Inc., Bethesda, MD, USA).

### 2.14. Statistical Analysis

Data were expressed as the mean ± standard deviation. Differences between groups were analyzed using the unpaired *t*-test. Differences across more than three groups were analyzed using the one-way Analysis of Variance (ANOVA) test with Holm-Sidak correction. GraphPad Prism software (version 6; GraphPad Prism Software Inc., San Diego, CA, USA) was used for the statistical analysis. The difference with a *p* value of <0.05 was considered statistically significant.

## 3. Results

### 3.1. Characterization of Self-Assembled Nanocomposites

The MALDI-TOF IMS characterization showed that the molecular weights of polyplexes R9, R9-N3, and CH3-R9-RGD were 3958.470 *m/z*, 1890.935 *m/z*, and 2558.000 *m/z*, respectively (Figure 2A). Gel electrophoresis showed that the amount of free siP3H4 was significantly decreased with the Nano@siP3H4 molar ratios increased (Figure 2B). Notably, free siP3H4 disappeared when the ratio reached 20/1, suggesting the complete encapsulation of free siP3H4 oligonucleotides by the CH3-R9-RGD nanoparticles. The DLS analysis showed that the size of polyplex nanoparticles was increased with Nano@siP3H4 ratio, without significant difference (Figure 2C). All complexes showed positive Zeta potentials when the Nano@siP3H4 ratios changed from 0/1 to 40/1. When the ratio reached 30/1, the particle diameter gradually stabilized at about 170 nm and the corresponding Zeta potential increased with the Nano@siP3H4 ratios. The DLS analysis showed that the complex had good stability in serum (Figure 2D,E). The size and PDI of the Nano@siP3H4 nanocomposites had no significant changes during the 48 h-incubation in serum (Figure 2D,E). The PDI was increased from 0.18 ± 0.01 at baseline to 0.20 ± 0.02 at 48 h (Figure 2E). TEM analysis showed that the size of nanoparticles (30/1) was about 160 nm, with a uniform spherical structure (Figure 2F).

### 3.2. In Vitro Cellular Uptake and Localization of Self-Assembled Nanocomposites

The analyses of cellular uptake and localization of self-assembled nanocomposites was performed in the T24 cells. 

LSCM showed that c(RGDfK) had a higher permeability than c(RADfK) (Figure 3A). Both were located in the cytoplasm. LSCM tracked the fluorescence signals of FAM-labeled siP3H4 in the cytoplasm of T24 cells with CH3-R9-RGD particles (Figure 3B). LSCM showed that CH3-R9-RGD nanoparticles (for 24 h) enhanced the permeability of ce6 in T24 cells as the fluorescence intensity of ce6 signal in cells treated with Nano@ce6/siP3H4 was significantly higher than that in cells treated with free ce6 (*p* = 0.0003, Figure 3C,D). In addition, LSCM analysis showed the co-localization of ce6 with FAM-labeled siP3H4 in the cytoplasm, showing the successful encapsulation and delivery of free siP3H4 and ce6 by the CH3-R9-RGD nanoparticles (Figure 3E). 

We also analyzed the endocytosis mechanism of self-assembled nanocomposites by treating T24 cells with endocytosis inhibitors CPZ (10 μg/mL), wortmannin (50 nM), MβCD (50 μM), and cRGD (100 μM) under normal (37 °C) or low temperature (4 °C). Flow cytometry analysis showed that low temperature, c(RGDfK), and the endocytosis inhibitors blocked the uptake of Nano@ce6/siP3H4 nanocomposites (Figure 3F). 

### 3.3. P3H4 Expression Was Influenced by ce6, siP3H4, and Laser Light

Before determining the cytotoxicity of self-assembled nanocomposites, we examined the expression levels of P3H4 mRNA and protein in T24 cells treated with Nano@ce6/siNC, Nano@siP3H4, and Nano@ce6/siP3H4 with and without laser light. PCR analysis showed that the level of P3H4 mRNA was significantly decreased by Nano@ce6/siP3H4 (*p* = 0.0015) and Nano@ce6/siP3H4 plus laser (*p* < 0.0001, Figure 4A). There was no significant difference in the level of P3H4 mRNA between other treatments due to the large standard deviation range. Western blot analysis indicated that Nano@siP3H4 and Nano@ce6/siP3H4 plus laser light decreased P3H4 protein expression (Nano@ce6/siNC vs. Nano@ce6/siNC laser, *p* = 0.0503; Nano@ce6/siNC vs. Nano@siP3H4, *p* = 0.0111; and Nano@ce6/siNC laser vs. Nano@ce6/siP3H4 laser, *p* = 0.0045; Figure 4B).

### 3.4. Exposure of CRT upon the Delivery of siP3H4 via Self-Assembled Photosensitive Nanocomposites

The exposure of CRT on the surface of the plasma membrane was detected using LSCM.

We found that laser light enhanced CRT exposure on the surface of the plasma membrane in T24 cells (Figure 4C,D). In addition, the encapsulation of ce6 by the CH3-R9-RGD polyplex enhanced laser light-induced CRT exposure, which was further enhanced by the co-delivery of ce6 and siP3H4 via the CH3-R9-RGD polyplex (Figure 4C,D). The CRT exposure upon ce6 and siP3H4 co-delivery in T24 cells showed that CH3-R9-RGD polyplex-mediated therapy for BC was related to ICD.

### 3.5. In Vitro Phototoxicity of Self-Assembled Nanocomposites

The in vitro cytotoxicity of the self-assembled siP3H4 delivery system was assessed by examining the cell viability and apoptosis in T24 cells. We found that cells treated with free ce6 with and without laser were non-cytotoxic to T24 cells (Figure 5A). Nano@ce6/siP3H4 with and without laser decreased T24 cell viability compared with cells treated with ce6 (Figure 5A). The cytotoxicity of Nano@ce6/siP3H4 nanocomposites was photosensitive and ce6 dose dependent. Notably, the phototoxicity of ce6 was dose-dependent at all test concentrations in cells treated with Nano@ce6/siP3H4 plus laser, but not for cells treated with Nano@ce6/siP3H4 minus laser.

In addition, we examined the ^1^O_2_ generation to evaluate the oxidative damage induced by nanocomposites. The results of LSCM analysis showed that laser light triggered ^1^O_2_ generation in T24 cells (*p* < 0.0001, Figure 5B). Flow cytometry showed that nanocomposites Nano@ce6/siNC, Nano@siP3H4, and Nano@ce6/siP3H4 promoted cell apoptosis (Figure 5C,D). We observed that laser light enhanced cell apoptosis in cells treated with Nano@ce6/siNC and Nano@ce6/siP3H4 (Nano@ce6/siNC vs. Nano@ce6/siNC laser, *p* = 0.0025; and Nano@ce6/siP3H4 vs. Nano@ce6/siP3H4 laser, *p* < 0.0001), but not in cells treated with Nano@siP3H4 (Nano@siP3H4 vs. Nano@siP3H4 laser, *p* > 0.05; Figure 5C,D). Notably, Nano@ce6/siP3H4 plus laser showed the highest efficiency in promoting cell apoptosis. 

### 3.6. Targeted Delivery and Biodistribution of Nanocomposites

The BC mouse tumor model was established (intravesical inoculation of MB49 cells, 1 × 10^6^ cells) to evaluate the tumor-suppressive effect of nanocomposites Nano@ce6/siNC, Nano@siP3H4, and Nano@ce6/siP3H4 (30/1) in vivo. The time-dependent distribution of Nano@ce6/siP3H4 using the in vivo imaging system showed that the nanocomposites were effectively accumulated in the liver and tumor sites. In addition, the fluorescent signals were retained for <48 h in tumor sites (Figure 6A,B). Provably, the administration of self-assembled nanocomposites Nano@ce6/siNC (vs. control, not significant), Nano@siP3H4 (*p* < 0.0001 vs. control), and Nano@ce6/siP3H4 (*p* < 0.0001 vs. control) suppressed the growth of in vivo BC tumors (Figure 6C). As expected, the irradiation of laser light enhanced the anti-tumor efficiency of Nano@ce6/siNC or Nano@ce6/siP3H4 (*p* < 0.0001) but not of Nano@siP3H4 (not significant, Figure 6C). Notably, the co-delivery of ce6 and siP3H4 by the CH3-R9-RGD polyplex (Nano@ce6/siP3H4 laser) had the highest efficiency in inhibiting tumor growth in vivo.

### 3.7. Therapeutic Function of Nanocomposites

H&E staining showed that most of the tumor cells in the control groups were normal in shape, without obvious cell apoptosis or necrosis (Figure 6D). Compared with control tumors, those tumors treated with nanocomposites Nano@ce6/siNC, Nano@siP3H4, and Nano@ce6/siP3H4 had severe tumor cell apoptosis or necrosis, with broken nucleus and diffused cytoplasm, especially in tumors treated with Nano@ce6/siP3H4 plus laser. Moreover, immunohistochemical analysis showed that those nanocomposites decreased Ki-67 protein expression in the nuclei of tumor cells (Figure 7A,B). Western blot analysis confirmed that nanocomposites Nano@siP3H4 and Nano@ce6/siP3H4 increased Caspase 3 expression and decreased P3H4 expression in BC tumor tissues (Figure 7C,D). Notably, nanocomposites Nano@ce6/siP3H4 plus laser light had the highest efficiency in reducing Ki-67 and P3H4 expression and in increasing Caspase 3 expression. 

In addition, flow cytometry showed that the fluorescence intensity of Cy3-labeled CRT in tumor cells was increased by nanocomposites Nano@ce6/siNC, Nano@siP3H4, Nano@ce6/siP3H4, and Nano@ce6/siP3H4 plus laser (Figure 8A,B). Nanocomposites Nano@ce6/siP3H4 with laser had the highest efficiency in promoting CRT exposure in tumor cells (Figure 8A,B).

## 4. Discussion

Great advances had been made in nanotechnological drug-delivery systems for anti-tumor application [29,30]. A large number of studies have shown that nanoparticles encapsulating siRNA and miRNAs are effective anti-tumor therapies for human cancers [9,13,31,32]. Efficient targeted systemic delivery of siRNA to tumor cells via cyclic RGD-installed nanoparticles is one of the most researched topics [11,18,29,33,34]. This study synthesized the BC tumor cell-targeting nanoparticles CH3-R9-RGD for the efficient delivery of siP3H4. The co-delivery of siP3H4 and photosensitizer ce6 through the CH3-R9-RGD nanoparticles to BC tumor cells showed higher anti-tumor efficiency than CH3-R9-RGD nanoparticles only encapsulating siP3H4. Experiments on BC tumor models showed that the self-assembled photosensitive nanocomposites CH3-R9-RGD@ce6/siP3H4 in a molar ratio of 30/1 inhibited P3H4 (mRNA and protein) expression, promoted CRT protein exposure, suppressed BC tumor cell proliferation, reduced ^1^O_2_ production, and promoted cell apoptosis in vitro and suppressed BC tumor growth in vivo. These results showed that the nanotechnological drug-delivery systems CH3-R9-RGD@ce6/siP3H4 had high tumor cell-killing efficiency in BC tumors.

The major advantages of nanocarriers for targeted drug delivery include good control over size, good biocompatibility, specific fluorescence, tumor cell targeting, security, and enhanced therapeutic efficacy [9,13,30,34]. Most nanosized nanoparticles have programmable and transmembrane sizes ranging from 30 to 350 nm, with positive Zeta potentials and remarkable fluorescent labeling [9,13]. The photosensitizer molecule ce6 is widely used in the synthesis of nanoparticles for photodynamic imaging and therapy because of its biomedical advantages, such as red-light fluorescence (maximum excitation wavelength = 660 nm) and non-toxicity [9,13,35,36,37]. The self-assembled nanocomposites CH3-R9-RGD@ce6/siP3H4 had a diameter of ~160 nm and a Zeta potential of +18 mV at the optimum molar ratio of 30/1, showing that CH3-R9-RGD nanocarriers had high ce6/siP3H4-loading capacity and dispersibility in tumor cells. The main component of the nanocarrier designed and synthesized in this study was a linear polypeptide R9 that had good water solubility [13]. The R9 peptide was used to conjugate ce6 to improve its solubility and biological compatibility. c(RGDfK) was modified by azide and installed on cationic polypeptide R9 to construct a water-soluble and remarkable tumor-targeting drug delivery system. Importantly, CH3-R9-RGD@ce6 had good permeability and triggered the generation of intracellular ROS (^1^O_2_) in T24 cells. Fluorescence analysis showed that CH3-R9-RGD@ce6 nanocomposites with laser light irradiation promoted the exposure of CRT on the surface of the plasma membrane in BC T24 cells, which was enhanced by the co-delivery of siP3H4. The results indicated that the self-assembled nanocomposites encapsulating siP3H4 and ce6-mediated combination immunotherapy.

RNAi is an effective regulation control for posttranscriptional gene expression. However, the application of siRNA therapy is hampered because of the lack of targeting ability. Recently, tumor-targeting nanoparticles were designed to improve the targeting ability and intracellular translocation efficiency of siRNA [29]. Research on nanotechnology-based systemic tumor-targeting delivery of siRNA against oncogenic genes is in full swing [17,29,38]. The delivery of siRNA to tumor cells via cyclic RGD-installed nanoparticles is an efficient and widely used tumor-targeting system [11,29,33,34]. The self-assembled tumor-targeting nanoparticles have effective tumor-specific distribution and accumulation [9]. However, one of the biggest challenges for the delivery of tumor-targeting nanoparticles is preventing them from accumulating in the liver or spleen [9,39]. Tsoi et al. [39] confirmed that poly(ethylene glycol) (PEG) was mainly taken by Kupffer cells, hepatic B cells, and liver sinusoidal endothelial cells in the liver. Tsoi et al. showed that the accumulation of nanoparticles in the liver is nanosize-dependent: the larger the particle, the more likely it is to be absorbed by the liver [39]. However, studies of cyclic RGD-installed nanoparticles, including 5-FU@SF-cRGDfK-Ce6: 365 nm [11], LCCN-Poly(I:C): 110 nm [9], and Ce6-R9-^125^I-RGD: 200 nm [13], showed that these nanoparticles were primarily accumulated in tumors, followed by in the skin, liver, or spleen [11]. We found that CH3-R9-RGD@ce6 nanoparticles were primarily accumulated in the liver and skin in normal mice. In BC tumor model mice, nanoparticles were accumulated in the skin, liver, and tumor sites. According to these studies, the accumulation of nanoparticles in the liver is material- and size-dependent.

SiRNAs encapsulated by tumor-targeting nanoparticles are released in the cytoplasm as the nanoparticles are broken down after entering tumor cells [29,40]. Lobovkina et al. [40] reported that the tristearin solid lipid nanoparticles had a prolonged siRNA release period of longer than 7 days. Most nanoparticles biodegradable materials, including PEG, polyethylenimine, peptide, poly(lactic-co-glycolic acid), polyinosinic-polycytidylic acid, and collagen, have a siRNA release period of 48–80 h in vivo [9,11,41,42]. The time-dependent in vivo biodistribution analysis of nanoparticles showed that CH3-R9-RGD@ce6/siP3H4 nanocomposites were released and broken down completely at 24–48 h post injection in normal mice. In BC tumor model mice, nanocomposites in tumor sites were broken down completely at 24–48 h post injection, and the breakdown of nanocomposites accumulated in the liver needed more than 48 h. These results indicate that the self-assembled CH3-R9-RGD@ce6/siP3H4 nanocomposites have good biodegradability.

By comparing the efficiency of nanocomposites, we found that CH3-R9-RGD@ce6/siP3H4 had the highest efficiency in tumor-targeting therapy. Mice in the CH3-R9-RGD@ce6/siP3H4 group had smaller tumor sizes than mice in the CH3-R9-RGD@ce6/siNC and CH3-R9-RGD@siP3H4 groups, suggesting that the efficient co-delivery of siP3H4 and ce6 by cRGDfK-installed tumor-targeting nanoparticles facilitated PDT for BC. As verified by our previous experiments, siP3H4 transfection into BC cancer cells (T24 and EJ cells) in vitro inhibited cell proliferation, migration, and invasion, and the administration of siP3H4 into BC tumors in vivo suppressed tumor growth [21]. This study showed that the delivery of siP3H4 via tumor-targeting nanoparticles into T24 cells in vitro reduced cell viability and promoted ROS generation, cell apoptosis, and CRT exposure. The CRT exposure upon co-delivery of ce6 and siP3H4 in T24 cells indicated that PDT for BC was related to ICD. Collectively, the self-assembled nanocomposites encapsulating siP3H4 and ce6 might be an alternative strategy for BC therapy or intravesical instillation chemotherapy.

## 5. Conclusions

In summary, this study synthesized a reliable and biocompatible tumor-targeting nanoparticle, a siP3H4 delivery system based on the assembly of ce6 and cRGDfK-modified R9 peptide. Experiments showed that the self-assembled CH3-R9-RGD nanoparticles encapsulating ce6 and siP3H4 were efficient for the tumor-targeting therapy for BC. CH3-R9-RGD@ce6/siP3H4 nanocomposites decreased cell proliferation and promoted cell apoptosis in T24 cells in vitro and suppressed tumor growth in vivo. In addition, laser light (660 nm) improved the anti-tumor effect of CH3-R9-RGD@ce6/siP3H4 nanocomposites. The CH3-R9-RGD nanoparticles encapsulating siP3H4 and ce6 might be an alternative strategy for BC PDT or at least efficient for intravesical instillation chemotherapy. However, more preclinical experiments on animal models should be performed to confirm the anti-tumor effect efficiency of the self-assembled nanocomposites.

## Figures and Tables

**Figure 1 pharmaceutics-14-02149-f001:**
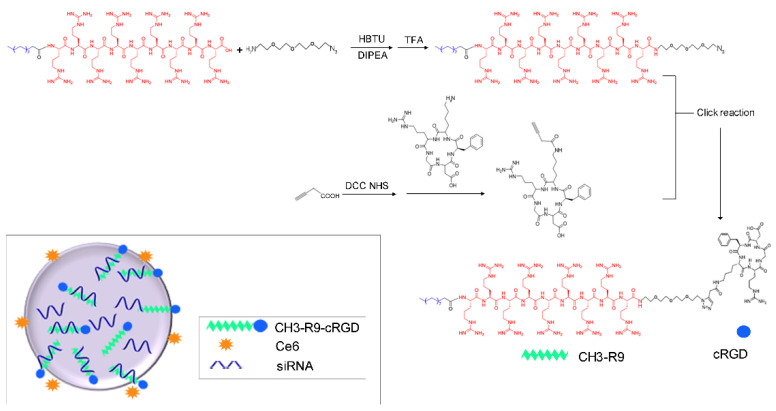
The schematic diagram showing the syntheses of the R9-N3 polymer and the CH3-R9-RGD(Nano)@ce6/siP3H4 nanocomposite. DIPEA, N,N-diisopropylethylamine. HBTU, O-(1H-benzotriazol-1-yl)-N,N,N′,N′-tetramethyluronium hexafluorophosphate. NHS, N-hydroxysuccinimide. TFA, trifluoroacetic acid.

**Figure 2 pharmaceutics-14-02149-f002:**
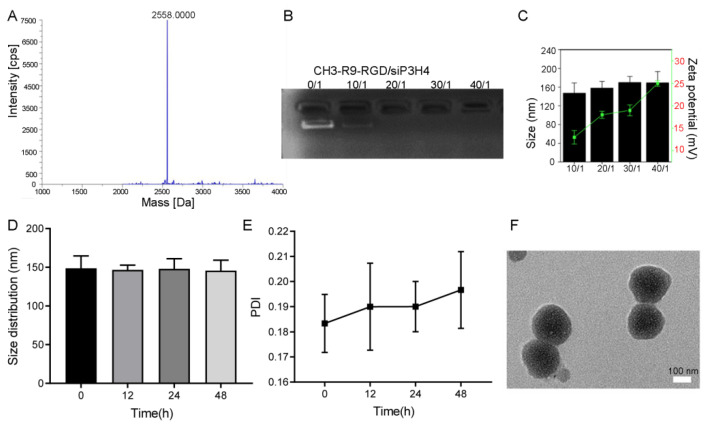
Characterization of the CH3-R9-RGD polyplex nanoparticles and the mixtures of CH3-R9-RGD (Nano)@siP3H4. (**A**): Fluorescence images of T24 cells cultured in medium containing c(RGDfK) and c(RADfK) (Cy3-labeled). (**B**): The identification of CH3-R9-RGD polyplex using the matrix-assisted laser desorption ionization-time-of-flight imaging mass spectrometry, with a mass of 2558.000 Da (the highest peak). (**C**): Gel electrophoresis analysis of the CH3-R9-RGD polyplex and free siP3H4 mixture with molar ratios at 0/1, 10/1, 20/1, 30/1, and 40/1. (**D**,**E**): Hydrodynamic sizes and polydispersity index (PDI) of the particles of CH3-R9-RGD polyplex with free siP3H4, respectively. (**F**) the transmission electronic image of the CH3-R9-RGD polyplex with siP3H4 (30/1). Scale bar = 100 nm. Data are shown as mean ± standard deviation.

**Figure 3 pharmaceutics-14-02149-f003:**
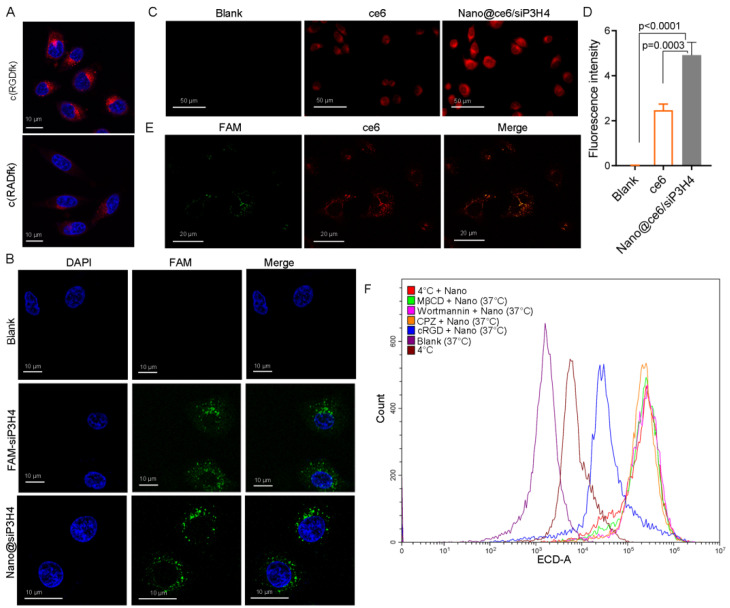
Laser scanning confocal microscopy (LSCM) of T24 cells cultured with self-assembled nanocomposites, siP3H4, and chlorin e6 (ce6). (**A**): The confocal fluorescence images of T24 cells cultured in medium containing c(RGDfK) and c(RGDfK). Scale bar = 10 μm. (**B**): The confocal fluorescence images of T24 cells cultured in FAM-labeled siP3H4 with and without CH3-R9-RGD(Nano) polyplex. The nuclear reagent is DAPI. Cells were treated with DAPI was used as blank control. Scale bar = 10 μm. (**C**,**D**): LSCM images and fluorescence intensity of T24 cells co-cultured with CH3-R9-RGD and FAM-labeled siP3H4 (30/1). Scale bar = 50 μm. (**E**): LSCM images showing the co-localization of ce6 and FAM-labeled siP3H4 in T24 cells. Scale bar = 20 μm. (**F**) flow cytometry analysis of FAM-positive T24 cells post treatments with self-assembled nanocomparticle CH3-R9-RGD with chlorpromazine (CPZ; 10 μg/mL), wortmannin (50 nM), methyl-beta-cyclodextrin (MβCD; 50 μM), and cRGD (100 μM) under normal (37 °C) or low temperature (4 °C).

**Figure 4 pharmaceutics-14-02149-f004:**
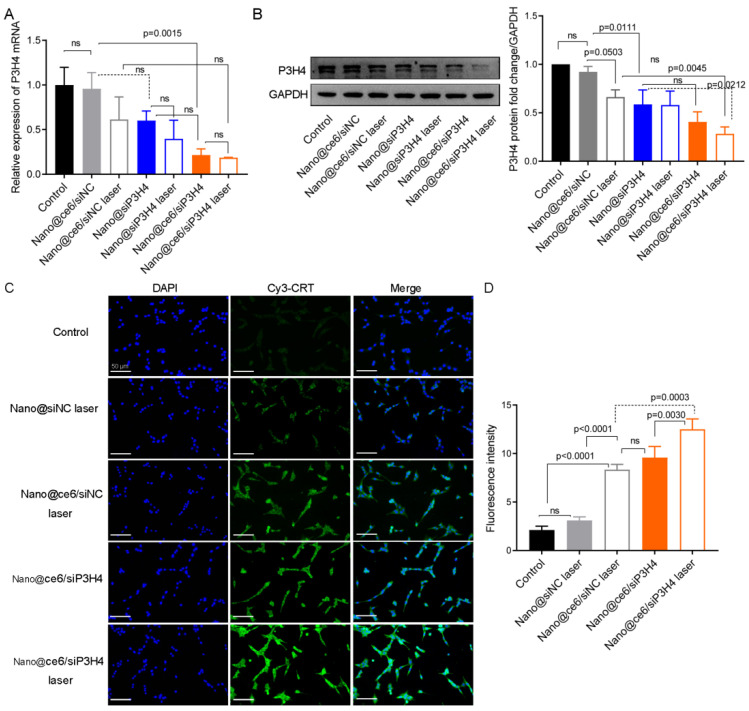
Effect of nanoparticles, ce6, and laser light on P3H4 expression and calreticulin (CRT) exposure in T24 cells. (**A**,**B**): The expression levels of P3H4 mRNA and protein in T24 cells cultured with self-assembled CH3-R9-RGD polyplex encapsulating siP3H4 and chlorin e6 (ce6), with and without laser light irradiation (660 nm for 1 min, followed by a 4 h-incubation). (**C**,**D**): Fluorescence images and intensity of T24 cells indicating the exposure of CRT (Cy3-labeled) on the surface of the plasma membrane, respectively. Scale bar = 50 μm. Data are shown as mean ± standard deviation. ns, not significant.

**Figure 5 pharmaceutics-14-02149-f005:**
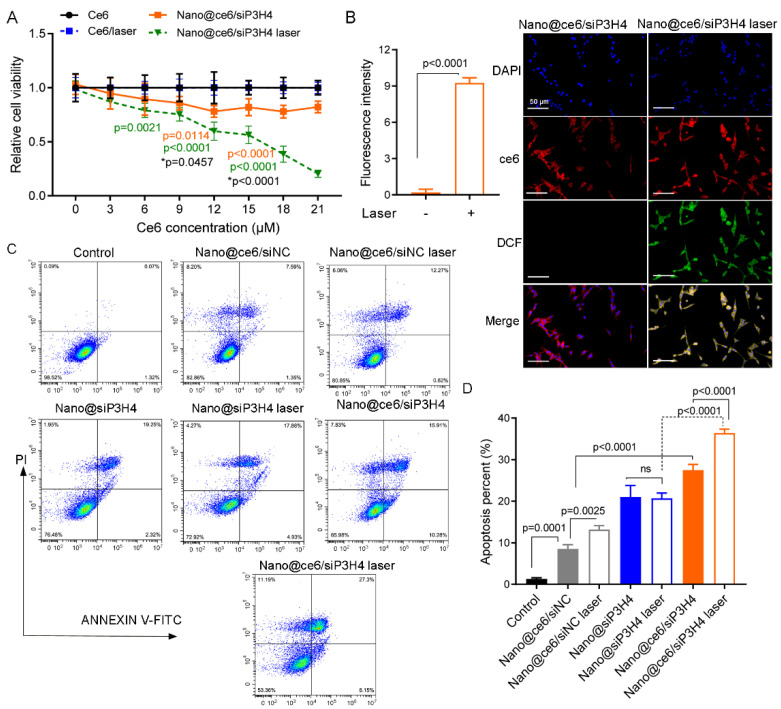
Study of in vitro cytotoxicity of self-assembled nanocomposites CH3-R9-RGD (Nano)@ce6/siNC, Nano@siP3H4, and Nano@ce6/siP3H4. (**A**): Cell viability of T24 cells cultured with chlorin e6 (ce6) and siP3H4 encapsulated by CH3-R9-RGD delivery system with and without laser light (660 nm for 1 min). The *p* value marked in green represent the difference between ce6/laser and CH3-R9-RGD(Nano)@ce6/siP3H4/laser, and the *p* value marked in orange represent the difference between ce6 and Nano@ce6/siP3H4. The *p* value marked by star (**p*) represent the difference between Nano@ce6/siP3H4 laser and Nano@ce6/siP3H4. (**B**): Fluorescence images showing the generation of intracellular reactive oxygen species (^1^O_2_) in T24 cells (by fluorogenic probe DCFH-DA). Cells were treated with Nano@ce6/siP3H4 and Nano@ce6/siP3H4 plus laser light (660 nm for 1 min) for 4 h. Scale bar = 50 μm. (**C**,**D**): Cell apoptosis analyzed by flow cytometry. Cells were treated with CH3-R9-RGD nanocomposites for 12 h, followed by irradiation (660 nm for 1 min) and a 36 h incubation. ns, not significant.

**Figure 6 pharmaceutics-14-02149-f006:**
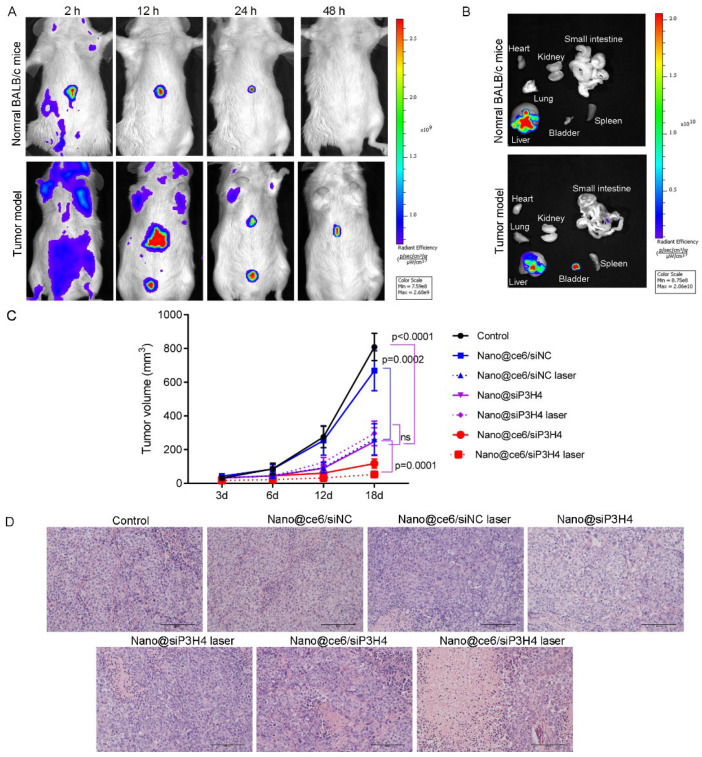
Targeted delivery and therapeutic function of nanocomposites for bladder cancer in vivo. (**A**): In vivo time-dependent fluorescence imaging of tumor-bearing mice at 2, 12, 24, and 48 h post tail vein injection of CH3-R9-RGD nanoparticles (Nano) encapsulated chlorin e6 (ce6) and siP3H4, followed by laser light (660 nm, 0.45 W/cm^2^ for 5 min). (**B**): Ex vivo fluorescence images of major organs and tumors. (**C**): In vivo tumor growth curve (*n* = 6 in each group). Tumor volume was calculated using the equation of volume = (L × W^2^)/2, where L and W are the larger and shorter width, respectively. Data are shown as mean ± standard deviation. (**D**): The representative images showing the hematoxylin and eosin (H&E) staining of tumor tissue sections from tumor-bearing mice (*n* = 5 in each group) after treatment. Scale bar = 400 μm. ns, not significant.

**Figure 7 pharmaceutics-14-02149-f007:**
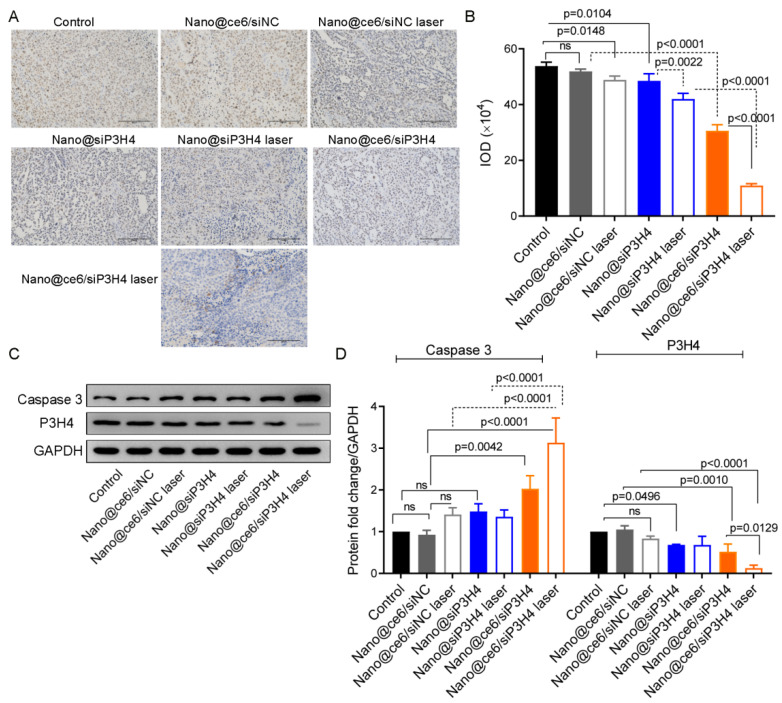
Ki-67, Caspase 3, and P3H4 proteins expression in tumor sections in vivo treated with nanocomposites. (**A**,**B**): Immunohistochemical analysis of Ki-67 protein in nuclei of tumors in vivo (*n* = 5 in each group). Scale bar = 400 μm. IOD, integrated optical density. (**C**,**D**): Western blot analysis of caspase 3 and P3H4 proteins in tumor tissues (*n* = 6 in each group). Data are shown as mean ± standard deviation. Mice were treated with CH3-R9-RGD nanoparticles encapsulated chlorin e6 (ce6) and siP3H4 or siNC, with or without laser light, for 20 days. ns, not significant.

**Figure 8 pharmaceutics-14-02149-f008:**
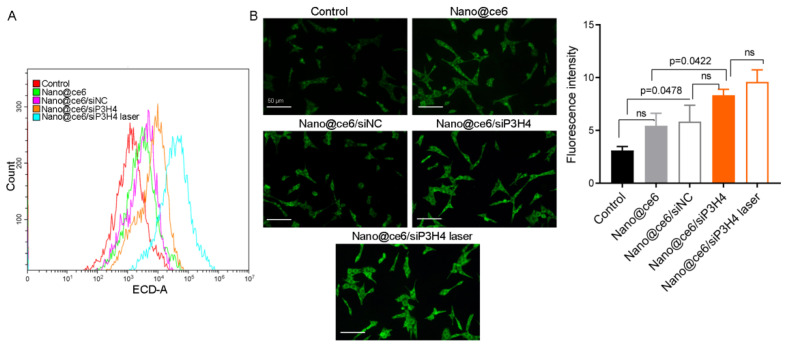
Exposure of calreticulin (CRT) protein in bladder cancer tumor cells from tumor-bearing mice. (**A**,**B**): Flow cytometry and fluorescent images of CRT exposure (Cy3-labeled) on the surface of the plasma membranes in tumor cells from tumor-bearing mice (*n* = 6 in each group). Mice were treated with self-assembled nanocomposites for 20 days and laser light (660 nm, 0.45 W/cm^2^ for 5 min). Scale bar = 50 μm. ns, not significant.

## Data Availability

The raw data are available from the corresponding author via hanchdoctor@st.btbu.edu.cn.

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
