# Peer review of "Efficient Delivery of P3H4 siRNA and Chlorin e6 by cRGDfK-Installed Polyarginine Nanoparticles for Tumor-Targeting Therapy of Bladder Cancer"

_pharmaceutics, 2022, doi:10.3390/pharmaceutics14102149_

Round 1

Reviewer 1 Report

The manuscript describes the preparation of a drug-conjugated gold nanoparticle system, and the evaluation of its delivery efficiency and anti-cancer capacity against bladder cancer. However, before making it into the final publication, certain doughs must be clarified to improve the present manuscript scientifically. Specific comments are as follows;

Major comments:

1.     Please make the labeling in each group to be more clear and enlarge in all results. Please provide the scale bar in all results.

2.     The quality in each results representation seems not good and poor, it’s necessary for a major modification in each image and labeling then make it to be more scientific sound for this manuscript.

Specific comments:

1.     Please give more background introduction for another nanocarrier for treatment with bladder cancer in the “Introduction” section.

2.     Please provide the cell viability test for normal cells to confirm this NP without any cell toxicity effect to avoid standard effect if it will be apply for clinical treatment application.

3.     Please explain why there were no dose-dependent manner for 20/1, 30/1, 40/1  in the Figure 1B?

4.     Please remove 50/1 in the “Figure 1C”

5.     Please included CH3 and CH3-R9 in the Figure 1D.

6.     Figure 1D (y-axis should be PDI rather than size distribution (nm) was wrongly representation and not consistent to Figure caption (zeta potential), please carefully revised this data.

7.     Why choice 30/1 rather than 10/1 in the Figure 1E? There are not consistent in the Figure 1B. Please explain and revise this data.

8.     Please provide the CH3 and CH3-R9 TEM image in the Figure 1E then make it to be more convince.

9.     Figure 2 data represent seems not good, please following suggestion as list below:

(1)  Please make the labeling to be more clear (Figure 2A-E)

(2)  Picture size to be consistent

y-axis should be MFI (mean fluorescent intensity rather than Count), please correct and provide quantification data.

(3)  Please provide scale bar in each result.

10.  “Flow cytometry analysis showed that low temperature and cRGD blocked the uptake of siP3H4 and CH3-R9-RGD nanocomposites, while endocyto sis inhibitors did not block it (Figure 2E). (line 319-320) Why endocytosis did not block from this result? It’s seems not make sense. Do you also check other inhibitors (ex: Baf, Cyto D).

11.  Please provide the quantification data of DCF-Da stating using flow cytometry in the Figure 4B as well as more detail explain why test ROS generation in this result.

12.  Please provide the quantification in the Figure 5D as well as sample number should addressed in the Figure caption.  

13.  Please provide the quantification in the Figure 6A as well as sample number should addressed in the Figure caption. 

14.  Please provide the quantification in the Figure 7B as well as sample number should addressed in the Figure caption. 

Author Response

Dear Pharmaceutics and Reviewers,

We sincerely appreciate the Pharmaceutics Editorial Office and reviewers for the review and comments on our work. The manuscript has been revised and necessary corrections and changes have been made according to the suggestions and comments. Please review the revised manuscript and send back your further suggestions. We will spare no effort to improve this paper.

Thank you for your attention and patience!

Best,

Han CH

07 Step 2022

Comments and Suggestions for Authors

The manuscript describes the preparation of a drug-conjugated gold nanoparticle system, and the evaluation of its delivery efficiency and anti-cancer capacity against bladder cancer. However, before making it into the final publication, certain doughs must be clarified to improve the present manuscript scientifically. Specific comments are as follows:

Major comments:

  1. Please make the labeling in each group to be more clear and enlarge in all results. Please provide the scale bar in all results.

Response: Thanks for your kind suggestion. The labels of all figures had been edited as suggested and scale bar of the images had been added.

  1. The quality in each results representation seems not good and poor, it’s necessary for a major modification in each image and labeling then make it to be more scientific sound for this manuscript.

Response: Thanks for your kind suggestion again. All the figures are presented with a resolution of 300 dpi, and the labels of all revised figures had been edited.

Specific comments:

  1. Please give more background introduction for another nanocarrier for treatment with bladder cancer in the “Introduction” section.

Response: Thanks for your kind suggestion. The introduction for other nanocarriers (gold, liposomes, polymeric, peptide nanostructures) had been added as suggested.

  1. Please provide the cell viability test for normal cells to confirm this NP without any cell toxicity effect to avoid standard effect if it will be apply for clinical treatment application.

Response: Sorry, the cytotoxicity of the particle to normal cells was not detected, and this might be a limitation.

  1. Please explain why there were no dose-dependent manner for 20/1, 30/1, 40/1 in the Figure 1B?

Response: Done as suggested. Because all free siP3H4 oligonucleotides were encapsulated completely by the CH3-R9-RGD nanoparticles when the molar ratio increased to 20/1”.

  1. Please remove 50/1 in the “Figure 1C”

Response: Done as suggested.

  1. Please included CH3 and CH3-R9 in the Figure 1D.

Response: Sorry, the sizes of CH3 and CH3-R9 were not analyzed using the DLS.

  1. Figure 1D (y-axis should be PDI rather than size distribution (nm) was wrongly representation and not consistent to Figure caption (zeta potential), please carefully revised this data.

Response: Thanks for your kind suggestion. The PDI distribution of the nanoparticle had been added.

  1. Why choice 30/1 rather than 10/1 in the Figure 1E? There are not consistent in the Figure 1B. Please explain and revise this data.

Response: The ratio ≥ 20/1 showed the complete embedding of free siRNA by the nanoparticle, and 30/1 is reasonable. Also, when the ratio reached 30/1, the particle diameter gradually stabilized.

  1. Please provide the CH3 and CH3-R9 TEM image in the Figure 1E then make it to be more convince.

Response: Sorry, the sizes of CH3 and CH3-R9 were not analyzed using the TEM.

  1. Figure 2 data represent seems not good, please following suggestion as list below:

(1) Please make the labeling to be more clear (Figure 2A-E)

Response: Done as suggested.

(2) Picture size to be consistent y-axis should be MFI (mean fluorescent intensity rather than Count), please correct and provide quantification data.

Response: Done as suggested, Figure 2D had been revised and fluorescent intensity was added.

(3) Please provide scale bar in each result.

Response: Done as suggested.

  1. “Flow cytometry analysis showed that low temperature and cRGD blocked the uptake of siP3H4 and CH3-R9-RGD nanocomposites, while endocytosis inhibitors did not block it (Figure 2E). (line 319-320) Why endocytosis did not block from this result? It’s seems not make sense. Do you also check other inhibitors (ex: Baf, Cyto D).

Response: Sorry, there is an error. As shown in the Figure 3, endocytosis did block the uptake of siP3H4 and CH3-R9-RGD nanocomposites.

  1. Please provide the quantification data of DCF-Da stating using flow cytometry in the Figure 4B as well as more detail explain why test ROS generation in this result.

Response: The the quantification in the Figure 5B (new) was not added, also the reason why test ROS generation had been added in the related result text.

  1. Please provide the quantification in the Figure 5D as well as sample number should addressed in the Figure caption.

Response: The the quantification in the Figure 5D (Figure 6D now) was not added, as it was the HE result and picture presentation results will be more reasonable, and sample number had been addressed in the figure caption.

  1. Please provide the quantification in the Figure 6A as well as sample number should addressed in the Figure caption.

Response: The the quantification in the Figure 6A (Figure 7A now) had been added, and sample number had been addressed in the figure caption.

  1. Please provide the quantification in the Figure 7B as well as sample number should addressed in the Figure caption.

Response: The the quantification in the Figure 7B (Figure 8B now) had been added, and sample number had been addressed in the figure caption.

Reviewer 2 Report

Please improve figures and discuss more graphic presentation in results section.

Author Response

Dear Pharmaceutics and Reviewers,

We sincerely appreciate the Pharmaceutics Editorial Office and reviewers for the review and comments on our work. The manuscript has been revised and necessary corrections and changes have been made according to the suggestions and comments. Please review the revised manuscript, and, if required, will spare no effort to revise it according to further comments.

Comments and Suggestions for Authors

Please improve figures and discuss more graphic presentation in results section.

Responses: The figures’ quality has been improved and the results section has been revised with more graphic presentation. Also, the other sections of the work have been improved according to the suggestions and comments from you and the other two reviewers. Please review the revised manuscript and send back your further suggestions. We will spare no effort to improve this paper.

Thank you for your attention and patience!

Best,

Han CH

03 Step 2022

Reviewer 3 Report

Referee Report

Title: Efficient Delivery of P3H4 siRNA and Chlorin e6 by cRGDfK-Installed Polyarginine Nanoparticles for Tumor-Targeting Therapy of Bladder Cancer

Manuscript ID: pharmaceutics-1837388

By Joiner et al

Submitted to Pharmaceutics (ISSN 1999-4923)

Comments

This work reported results of cell and preclinical experiments on the delivery of P2H4 and polyarginine nanoparticles for tumour-targeting of bladder cancer. The manuscript is quite well-written and comprehensive. The study is topical and quite novel. I only have some minor comments for further improvements:

1.       Abstract, L21: Please provide the long form for BC (i.e. Bladder Cancer).

2.       Introduction, L38: There should be space between “the” and “12th”.

3.       Introduction, L57: To mention recent progress of nanotechnology in cancer therapy. Please consider to include some more updated references: Siddique et al (Nanomaterials 2020,10,1700), Siddique et al (App Sci 2020,10,3824) and Chow (Application of Nanoparticle Materials in Radiation Therapy. In Handbook of Ecomaterials. Leticia Myriam Torres Martinez, Oxana Vasilievna Kharissova and Boris Ildusovich Kharisov (Eds.), Springer Nature, Switzerland, Chapter 150, pp. 3661-3681, 2017.)

4.       Materials L89: Why the word “Fmoc-Arg-pbf-OH” is in red font?

5.       Section 2.2 and 2.3: It is good to have a schematic diagram to show the syntheses of the polymer (in 2.2) and nanocomposite (in 2.3).

6.       Figure 2: Caption: “Tem” should read “Temp”.

7.       Figure 3: There are two subfigures in B but one subfigure in A.

8.       Conclusions: Please mention what will be the next step after the cell and preclinical experiments in this work.

Author Response

Dear Pharmaceutics and Reviewers,

We sincerely appreciate the Pharmaceutics Editorial Office and reviewers for the review and comments on our work. The manuscript has been revised and necessary corrections and changes have been made according to the suggestions and comments. Please review the revised manuscript and send back your further suggestions. We will spare no effort to improve this paper.

Thank you for your attention and patience!

Best,

Han CH

02 Step 2022

Comments and Suggestions for Authors

Comments

This work reported results of cell and preclinical experiments on the delivery of P2H4 and polyarginine nanoparticles for tumour-targeting of bladder cancer. The manuscript is quite well-written and comprehensive. The study is topical and quite novel. I only have some minor comments for further improvements:

  1. Abstract, L21: Please provide the long form for BC (i.e. Bladder Cancer).

Response: Done as suggested: “Bladder cancer (BC)”.

  1. Introduction, L38: There should be space between “the” and “12th”.

Response: Done as suggested.

  1. Introduction, L57: To mention recent progress of nanotechnology in cancer therapy. Please consider to include some more updated references: Siddique et al (Nanomaterials 2020,10,1700), Siddique et al (App Sci 2020,10,3824) and Chow (Application of Nanoparticle Materials in Radiation Therapy. In Handbook of Ecomaterials. Leticia Myriam Torres Martinez, Oxana Vasilievna Kharissova and Boris Ildusovich Kharisov (Eds.), Springer Nature, Switzerland, Chapter 150, pp. 3661-3681, 2017.)

Response: Thanks for your recommendation and patience! These articles have been reviewed and cited them where appropriate, in the introduction and/or discussion section.

  1. Materials L89: Why the word “Fmoc-Arg-pbf-OH” is in red font?

Response: This was a clerical error, which has been corrected.

  1. Section 2.2 and 2.3: It is good to have a schematic diagram to show the syntheses of the polymer (in 2.2) and nanocomposite (in 2.3).

Response: Thanks for your kind suggestion. This figure is the latest version of Figure 1.

  1. Figure 2: Caption: “Tem” should read “Temp”.

Response: Done as suggested.

  1. Figure 3: There are two subfigures in B but one subfigure in A.

Response: Figure A is the result of PCR, while the results in Figure B are Western blot (gel plot is necessary).

  1. Conclusions: Please mention what will be the next step after the cell and preclinical experiments in this work.

Response: Thanks for your kind suggestion. The conclusion section has been revised and the next plan has been mentioned: More preclinical experiments on animal models should be extended and performed to confirm the anti-tumor effect efficiency of the self-assembled nanocomposites.”.

Round 2

Reviewer 1 Report

Journal:Pharmaceutics (ISSN 1999-4923)

Manuscript ID:pharmaceutics-1837388

Authors were adequately response all comments. However, there still a minor revision need completed list as below:

Minor comment: 

Please provide scale bar in the Figure 3A-D.

Author Response

Dear Pharmaceutics and Reviewers,

We sincerely appreciate the Pharmaceutics Editorial Office and reviewers for the review and comments on our work. The manuscript has been revised and necessary corrections and changes have been made according to the suggestions and comments. Please review the revised manuscript and send back your further suggestions. We will spare no effort to improve this paper.

Thank you for your attention and patience!

Best,

Han CH

28 Step 2022

Comments and Suggestions for Authors

Major comments:

  1. Please provide scale bar in the Figure 3A-D.

Response: The scale bars in Figure 3 had been provided in the first revision. Please check it.

Reviewer 3 Report

I am satisfied with the corrections and additional materials made by the authors as per my comments. The quality and presentation of this work are improved and I have no further questions.

Author Response

Dear Reviewer,

We sincerely appreciate your review and comments on our work, and the thank you for your approval of the revised files.

Best,

Han CH 

28 Step 2022